# Oxidative Stress Linking Obesity and Cancer: Is Obesity a ‘Radical Trigger’ to Cancer?

**DOI:** 10.3390/ijms24098452

**Published:** 2023-05-08

**Authors:** Mirna Jovanović, Sanja Kovačević, Jelena Brkljačić, Ana Djordjevic

**Affiliations:** Institute for Biological Research “Siniša Stanković”—National Institute of Republic of Serbia, University of Belgrade, 11060 Belgrade, Serbia; sanja.kovacevic@ibiss.bg.ac.rs (S.K.); brkljacic@ibiss.bg.ac.rs (J.B.); djordjevica@ibiss.bg.ac.rs (A.D.)

**Keywords:** obesity, cancer, oxidative stress

## Abstract

Obesity is on the rise worldwide, and consequently, obesity-related non-communicable diseases are as well. Nutritional overload induces metabolic adaptations in an attempt to restore the disturbed balance, and the byproducts of the mechanisms at hand include an increased generation of reactive species. Obesity-related oxidative stress causes damage to vulnerable systems and ultimately contributes to neoplastic transformation. Dysfunctional obese adipose tissue releases cytokines and induces changes in the cell microenvironment, promoting cell survival and progression of the transformed cancer cells. Other than the increased risk of cancer development, obese cancer patients experience higher mortality rates and reduced therapy efficiency as well. The fact that obesity is considered the second leading preventable cause of cancer prioritizes the research on the mechanisms connecting obesity to cancerogenesis and finding the solutions to break the link. Oxidative stress is integral at different stages of cancer development and advancement in obese patients. Hypocaloric, balanced nutrition, and structured physical activity are some tools for relieving this burden. However, the sensitivity of simultaneously treating cancer and obesity poses a challenge. Further research on the obesity–cancer liaison would offer new perspectives on prevention programs and treatment development.

## 1. Introduction

Body mass increase is mainly a result of overnutrition and sedentarism but can also be a result of genetic factors [1]. The energy surplus is stored in lipid depots, primarily in the fat tissue. Adults are classified as underweight, normal weight, overweight, or obese according to the body mass index (BMI), with BMI ≥ 25 delimitating overweight and BMI ≥ 30 obesity. Significantly, metabolic health goes beyond a change in total body mass and simplified parameters for its assessment; cardinal aspects to be considered in the context of health are body fat and muscle mass ratios, body fat distribution, the quality and balance in daily nutrients, and physical activity.

In 2016, 13% of the world’s adult population was estimated to be obese, whereas 39% were overweight [2]. The number of obese people is nearly three times higher than 50 years ago. Estimates made before 2020 were that in the majority of European countries, more than one in five people will be obese by 2025 [3], while in the USA, by 2030, every second adult will be obese [4]. These projections preceded and, to that end, could not foresee the yet-to-be-unraveled effects the coronavirus disease 2019 (COVID-19) pandemic left behind. An increase in appetite and weight gain have been considered one of the long-haul effects of COVID-19 [5]. While it is too early to give accurate numbers, a study published in 2022 reported an increase in average BMI by 0.6% and in obesity prevalence by 3% in the USA population during the period of lockdown in 2020 [6]. Both obesity and being overweight are major risk factors for non-communicable diseases (Figure 1), such as type 2 diabetes, coronary heart disease, stroke, as well as certain types of cancer, to name a few [7,8]. Despite extreme economic inequality, it is considered that today, obesity is causing more deaths worldwide than hunger in all parts of the world, apart from sub-Saharan Africa and Asia.

Cancer is a vast, heterogeneous group of diseases, with the major common traits being the independent growth and proliferation of cells that go beyond organism control. It can affect any organ in the human body. Although cancer types are vastly different, and even the cells within the same tumor might bare significant differences, the fundamental processes bringing them to existence can be relatively similar [9,10,11]. About half of the deaths in men, as well as almost 40% in women, are caused by cancers connected to modifiable risk factors that are considered preventable [12]. The leading preventable causes of cancer are considered to be smoking, alcohol use, being overweight, and obesity. The International Agency for Research of Cancer has identified 13 cancer hotspots where the disease is considered to be preventable by weight reduction in obese patients: esophagus adenocarcinoma, gastric cancer, colorectal, liver, pancreas, postmenopausal breast, endometrial, ovary, renal cell, meningioma, thyroid, gallbladder, and multiple myeloma [13]. Among them, breast, pancreas, colorectal, liver, and thyroid cancer are placed in the top 10 with the highest incidence and mortality in the world [14]. The number of metabolic diseases-contributed cancer deaths in the period between 2010 and 2019 had the sharpest incline compared to other modifiable factors. In fact, due to the decline of smoking, at the present rate, it is estimated by 2043, obesity will overtake smoking as the lead preventable cause of cancer in women in the UK [15]. 

Obesity can be a significant factor in cancer development (Figure 1). Recently, Harris et al. 2022 [16] systematically described how obesity affects cancer hallmarks. Here, we will look in more detail at how the oxidative stress that develops in obesity provides a “fertile ground” for cancer development and influences tumor progression. This review will briefly present the molecular connection between obesity and different types of cancer through oxidative stress. We propose a hypothesis that oxidative stress, consequential to metabolic disorder, and molecular pathological changes induced by constitutive high-caloric and/or high-fat intake in obese individuals play an important part in tumorigenesis and set a formidable milieu for cell transformation and malignancy progression. 

## 2. The “Goldilocks Paradox” of Reactive Species

Oxidative stress is a result of an imbalance between the concentration of reactive species and the efficiency of the systems—comprising enzymes and small molecules that mitigate them. Some examples of reactive oxygen species are superoxide (O_2_^·−^) hydrogen peroxide (H_2_O_2_), hydroxyl radical (^·^OH), ozone (O_3_), hypochlorous acid (HOCl), singlet oxygen (^1^O_2_), lipid (hydro)peroxides (LOOH), peroxyl radical (LOO^·^), and alkoxyl radical (RO^·^); relevant reactive nitrogen species are nitric oxide (^·^NO) and strong oxidizing agent peroxynitrite (ONOO^−^) [17]. Different reactive species interact, e.g., NO and O_2_^·−^ react to give ONOO^−^, and the acronym for reactive oxygen and nitrogen species (RONS) emerged as a description of their interrelationship. The deleterious effects of reactive species, i.e., damage to DNA, proteins, and lipids that ultimately lead to cell death, were first discovered, as was the obvious relationship they have with living systems. Over time and with advances in methodology, an important aspect of reactive species related to life emerged—namely, reactive species as mediators in cell signaling. However, to participate in cell signaling, they must meet several requirements: Their synthesis and clearance must be coordinated to maintain redox balance, and they must be stable enough to perform the required task. The two RONS that meet these conditions are H_2_O_2_ [18] and ^·^NO [19]. Both species mediate signal transduction by oxidative post-translational modifications of proteins that modify protein function [18,19,20,21].

When the cell is exposed to low-level H_2_O_2_, the immediate response is to shift glucose metabolism from glycolysis to the pentose phosphate pathway (PPP), which allows increased production of nicotinamide adenine dinucleotide phosphate (NADPH) [22]. ROS, in fact, causes NADPH deficiency because it is used as an electron donor for the detoxification systems (mainly glutathione and thioredoxin system) and is oxidized to NADP^+^. Increased NADP^+^ leads to higher activity of the major rate-limiting enzyme in PPP, glucose-6-phosphate dehydrogenase (G6PD), which determines the flux of G6P into this metabolic pathway [23]. At the same time, through ROS oxidation of their cysteine (Cys) residues, the proteins’ main redox molecular switches, glyceraldehyde-3-phosphate dehydrogenase (GAPDH) and pyruvate kinase M2 (PKM2), block glycolysis, ensuring the detour of glucose catabolism products toward PPP. In addition, several electron-transport chain (ETC) proteins are regulated through Cys residues as well [24]; thus, this immediate response to ROS is followed by a reduction in mitochondrial respiration to limit endogenous ROS production [25]. Adenosine monophosphate-activated protein kinase (AMPK) is activated to promote NADPH production and prevent anabolic processes that require NADPH consumption [26]. In summary, brief exposure to subtoxic concentrations of ROS leads to metabolic adaptations of the cell to avoid redox imbalance (Figure 2).

When the cell is exposed to increased RONS for a prolonged period of time, the genetically programmed response is initiated, and the detoxification systems are activated (Figure 2). The first line of defense is mainly controlled by nuclear factor E2-related factor 2 (Nrf2), the main redox balance regulator. By binding to the antioxidant response element (ARE 5′-TGACXXXGC-3′), Nrf2 promotes the expression of *PRDX*, *TRX1*, *TRXR1*, *GCL*, *GSR1*, *GPX*, and *CAT*, as well as about 200 other cytoprotective genes [27]. If the Nrf2-induced response still fails to handle the harmful RONS, other transcription factor genes get their chance to restore homeostases, such as activator protein 1 (AP-1), hypoxia-inducible factor 1α (HIF-1α), class O of the forkhead box transcription factors (FOXOs), nuclear factor kappa B (NFκB), and peroxisome proliferator-activated receptor-gamma coactivator 1α (PGC-1α). 

The major redox systems in the cell are the glutathione (GSH) and thioredoxin (Trx) systems. The GSH system consists of GSH, a small tripeptide that accepts electrons via the thiol group, the GSH-synthesizing enzyme glutamate-cysteine ligase (GCL), glutathione peroxidases (GPx)—a group of enzymes that catalyze the neutralization of H_2_O_2_ and lipid peroxides on account of GSH, and glutathione reductase (GR), reducing the oxidized glutathione (GSSG) on account of FAD group and NADPH. Components of the Trx system include Trx, a small redox protein with a thiol group that has numerous redox-regulated protein targets in the cell, and thioredoxin reductase (TrxR), whose main role is to reduce oxidized Trx at the expense of NADPH. Trx-interacting protein (TXNIP) is often assigned to this group because one of its roles is to inhibit Trx. These two thiol systems are the mainstays of redox homeostasis within the cell. Other enzymes and small molecules also contribute to the maintenance and regulation of free radical concentration and damage control made by RONS: superoxide dismutases (SODs) reduce the superoxide anion into less toxic H_2_O_2_, catalase (CAT) neutralizes H_2_O_2_, and peroxiredoxins (Prx), a group of enzymes, catalyze the reduction of H_2_O_2_, as well as organic hydroperoxides and peroxinitrates. Trx reduces oxidized Prx, thus enabling it to perpetuate the peroxide regulation. In addition, glutathione-S-transferases (GST) conjugate free radicals and xenobiotics to GSH, thus participating in the antioxidant defense of the cell as well. The key to redox balance maintenance is the fine-tuned cooperation between these antioxidant lines of defense and response regulated by the type and concentration of the radical species. As pointed out by Cheung and Vousden [10], SODs cannot tackle oxidative stress alone if there is not a “partner” waiting to take over the H_2_O_2_ produced. On the other hand, GR and TrxR are NADPH-dependent, meaning antioxidant detoxification systems must cooperate with metabolic pathways producing and relaying on NADPH [28].

RONS can induce damaging modifications to proteins and DNA, and oxidize lipids, creating new reactive species in the process [29]. If detoxification systems fail to protect the cell from the accumulation of damage caused by RONS, the cell triggers some of the self-destruction systems (apoptosis, necroptosis, ferroptosis, pyroptosis, autophagy, etc., depending on the context; Figure 2) [29]. For example, a substantial accumulation of reactive species activates p38/MAPK-dependent apoptosis. H_2_O_2_ inactivates PTEN, which is an inhibitor of the PI3K/Akt signaling pathway. Reduced Trx automatically inhibits the p38/MAPK pathway; being in the oxidized state due to RONS detoxification, it no longer inhibits the cell death signaling. 

As Goldilocks seeks the balance between too cold and too hot porridge in the fairytale, so do normal cells seek and maintain redox balance. Too little of RONS would mean a “communication break” since the free radical species have important roles in the cell signaling process and can act as efficient switches and regulators of effector proteins; too much of them cause irreparable damage to the building blocks and result in cell death. However, a perilous zone lurks between redox-balanced normal cells and cell death—cells that bare mutations and damage but manage to adjust and survive can obtain the transformed neoplastic phenotype and go rogue on the organism’s harmonious whole.

## 3. Metabolic Pathology in Obesity Brings Systemic Havoc

Obesity has become an inevitable burden of modern lifestyle that is characterized by high-fat, high-sugar intake through fast, processed food and drinks and an inactive way of life [30]. Most of the obesity treatments available today deal with visual and aesthetic consequences, but obesity is a problem far beyond that. Metabolic changes that happen as a result of physical inactivity and overload of high-calorie, nutrient-poor food are making an environment suitable for the development of life-threatening diseases such as stroke, type 2 diabetes, cardiovascular abnormalities, and cancer [31].

### 3.1. Metabolic Environment in Obesity

Increased energy intake is primarily addressed by the adipose tissue, the main tissue specialized for energy storage in the form of lipids. Nevertheless, adipose tissue is not just a simple energy depo—it is also a very important endocrine center that synthesizes and secretes hormones (adipokines) and communicates with the liver, gut, brain, etc., regulating energy homeostasis. Adipose tissue can adjust to long-term nutrient overload by expanding its capacity for lipid storage by increasing the size of the adipocytes (hypertrophy) and the number of differentiated cells in the tissue (hyperplasia) [32]. In obesity, the capacity of adipose tissue is surpassed, and lipids start to accumulate in surrounding tissues and organs that are not specialized for this function, such as the liver, skeletal muscle, and kidneys, causing lipotoxicity, inflammation, and systemic oxidative stress) [33]. Indeed, an increase in oxidative stress markers and impaired antioxidant defense was reported in different stages of obesity, during its development as well as in obese patients with and without insulin resistance [34]. Recently, it was shown that nitrosative stress and glycoxidation of proteins are increased in obese women compared to healthy controls as well as that bariatric surgery vastly normalizes detected abnormalities in these processes [35]. Interestingly, it has been shown that nutritionally related glycoxidation increases the tumorigenic potential of different cancer types [36,37] and glycoxidation has been recognized as a contributor to the vicious cycle in the development of lung cancer [38]. A tight inverse association was shown between the body fat percentage and antioxidant capacity [34] and decreased anti-oxidative defense, together with increased nitrosative stress, was found in patients with prostate and breast cancers [39]. At a systemic level, the presence of RONS and low-grade inflammation is recognized as the main characteristics of obesity [31,40].

Although adipose tissue is specialized for lipid storage, its long-term lipid overload also comes with the price of impaired adipokine secretion and insulin sensitivity, dyslipidemia, a rise of tissue inflammation, and oxidative stress (Figure 3), which all can stimulate biological mechanisms underlining cancer onset, progression, and metastasis [41]. Several studies, including ours, reported the presence of oxidative stress in the adipose tissue of animals genetically predisposed to be obese or fed a high-fructose diet [42,43]. One of the main enzymes that produce ROS outside of mitochondria is nicotinamide adenine dinucleotide phosphate oxidase (NOX). This enzyme uses intracellular NADPH as a donor of an electron in the process of production of O_2_^·−^ and H_2_O_2_ [44]. Obese mice were shown to have higher levels of NOX4 in adipose tissue [42], and a high-fat diet was followed by adipose tissue upregulation of NOX4 and ROS production [45]. Interestingly, overexpression of NOX proteins was detected in many malignancies. Inhibition of NOX4 has been shown to decrease the aggressiveness of non-small cell lung cancer, inhibit cell adhesion and invasive potential of gastric cancer, and suppress cell growth in human neuroblastoma cells [46,47,48]. Other isoforms of NOX are also involved in the onset and progression of different types of cancers: NOX2 was connected with the growth of prostate, colorectal, breast, and gastric tumors as well as myelomonocytic leukemia; NOX5 in Barrett’s esophageal adenocarcinoma and prostate cancer [48]. Pro-tumorigenic role of NOX proteins has led scientists to consider it as an important therapeutic target in cancer, but its obesity-related increase is shedding new light on its functions.

Obesity was shown to change the synthesis and secretion of adipokines (Figure 3), and shifts in adipokine regulation have been associated with different types of cancers and their metastatic potential [49]. Patients with esophageal and hepatocellular carcinomas have increased levels of leptin, known to stimulate growth hormones that can increase angiogenesis, cellular proliferation, and differentiation or inhibit apoptosis [50,51]. In line with this, adiponectin, which has an anti-inflammatory and protective role from cancerogenesis, was found to be reduced in several types of tumors, and its lower level in circulation was associated with cancer severity [52].

The enlargement of adipose tissue is followed by transient hypoxia and the rise of inflammation (Figure 3). Anti-inflammatory M2 macrophages are helping adipose tissue preserve its homeostasis through remodeling processes, but prolonged lipid overload can initiate a macrophage switch from anti- to pro-inflammatory, leading to increased production of pro-inflammatory cytokines IL-1β, IL-6, TNFα, etc. Prolonged and unresolved inflammation has been connected with DNA damage progressing into colon and liver cancer as well as premalignant mouth lesions [53,54,55,56,57]. In addition, macrophages can produce ROS, which may further potentiate the expression of pro-inflammatory adipokines (MCP-1, IL-6, PAI-1), decrease the expression of anti-inflammatory adipokines (adiponectin), and inhibit anti-oxidative enzymes (SOD1, SOD2, CAT, etc.) [42]. Different kinds of ROS can trigger various mechanisms to activate and sustain inflammation on their own. H_2_O_2_ was shown to activate the main pro-inflammatory transcriptional regulator, NFκB, by stimulating phosphorylation and dissociation of the NFκB inhibitor, IκB, from the NFκB-IκB complex [58]. Both inflammation and oxidative stress were shown to have the ability to cause insulin resistance. It has been shown by us and others that inflammation can, directly and indirectly, lead to insulin signaling impairment and resistance of adipose tissue cells to insulin [43,59,60]. Similarly, accumulated ROS, as well as proteins damaged by ROS, can activate c-Jun-N-terminal kinase (JNK) and cause inhibitory phosphorylation of insulin receptor substrate 1 (IRS1) on Ser307, resulting in insulin signaling impairment [34,58,61]. Both hyperinsulinemia and insulin resistance have been shown to decrease the levels of insulin-like growth factor-binding proteins (IGF-BPs), therefore, increasing the level of IGF-1 [51], which has been correlated with breast, pancreas, and colorectal cancer [62,63,64]. In addition, IGF-1 has been associated with a decline in globulin that binds and transports sex hormones. Therefore, an increase in free estrogen, which is known to upregulate genes involved in cell proliferation and progression of a cell cycle, could be the base of the connection between obesity and increased risk of breast and endometrial cancers [65,66]. This obesity-caused vicious cycle between inflammation and oxidative stress in the adipose tissue could be the driving force initiating adipose tissue metabolic disturbance, making specific molecular surroundings that can potentiate tumor onset and propagation.

This is why measuring levels of oxidative stress markers has been proposed to be an informative and important predictor of the onset and development of the main metabolic disturbances that accompany obesity, such as hypertension, insulin resistance, and the development of metabolic syndrome and cancer [67,68]. Interestingly, rats fed a high-fructose diet had increased levels of anti-oxidative enzymes and pro-inflammatory cytokines even before adipose tissue mass or BMI was changed, and obesity formally developed [43]. Similarly, an increase in adipose tissue depots in non-obese men was shown to be followed by elevated levels of lipid peroxidation [69]. The high-fat diet was followed by upregulation of NOX4 and ROS production in adipose tissue before the onset of obesity (or insulin resistance) [45]. This indicates that oxidative stress has the potential to be the cause and probable underlying mechanism of obesity development and the onset of associated metabolic disturbances in addition to its position as a metabolic consequence of obesity.

Off-topic, there are sex differences in the intensity of oxidative stress and anti-oxidative defense as well as in the incidence of different cancer types [70,71]. Some of the differences could be attributed to social and environmental factors (for example, increased use of tobacco in males compared to females in developed countries), but most disparities come from their different anatomy and physiology with an accent on the endocrine system and different roles of testosterone and estrogen [72]. Nevertheless, oxidative stress is the most common underlining mechanism of sex-specific carcinogens [70] (PMID: 27481070), which is why genders were not evaluated separately in this review and focus was given to the general molecular mechanism of oxidative stress in cancer and obesity.

### 3.2. Obesity-Induced Oxidative Stress Causes DNA Instability

Elevated oxidative stress in obesity can have direct and indirect effects on DNA stability and, consequently, tumorigenesis. Namely, when exposed to oxidative stress, DNA nucleotides can be oxidized as well. The most common oxidative changes in DNA made by reactive species are the 7,8-dihydro-8-oxoadenine and 8-hydroxy-2′-deoxyguanosine (8-OHdG). Guanines are considered the most vulnerable because, in comparison to other bases, they possess low redox potential [31]. Oxidized guanine bases can serve as a place for replication mistakes and substitutions. Namely, there is up to a 75% chance that DNA polymerase will, instead of cytosine, incorporate adenine opposite to oxidized guanine [73]. This can result in GC to TA mutation, but also GC to AT and even GC to CG [31,74,75]. Interestingly this is exactly the type of mutation frequently detected in the p53 tumor suppressor gene and RAS oncogene in patients with breast, lung, and skin cancers [76,77,78]. Beyond replication mistakes, the oxidation of guanines can influence surrounding nucleotide sequences and, for example, inhibit the interaction of methyl-cytosine and methyl-binding proteins changing epigenetic patterns [79]. There are some indications connecting the level of DNA oxidative damage and obesity. Measurement of 8-OHdG in urine and blood showed a positive correlation with the obesity status of adults, teenagers, and children and even the status of gestational diabetes mellitus in pregnant women [80,81,82]. By estimate, there is an increase of up to 50% of 8-OHdG in transformed cells compared to non-transformed cells [83], which is why 8-OHdG is considered a reliable marker of ROS-induced mutagenesis and tumorigenesis.

Apart from the direct effect that obesity-related ROS can have on DNA, studies show that oxidative stress caused by obesity can achieve indirect effects through byproducts of lipid peroxidation or secondary bile acid metabolites [31]. As mentioned, adipose tissue dysfunction is one of the hallmarks of obesity and related metabolic disorders. Obesity-initiated oxidative stress leads to increased lipid peroxidation in adipose tissue and the formation of reactive aldehyde byproducts malondialdehyde (MDA), 4-hydroxynonenal (4-HNE), and acrolein [84,85]. It has been shown that MDA can cause frameshift mutations and substitutions in sequences rich in repetitive CG [86]. Niedernhofer et al. [87] investigated the mutagenic effects of MDA on human cells, showing that MDA can induce DNA inter-strand crosslinks that can interfere with replication and transcription. If not repaired properly, DNA damage made by MDA can have detrimental effects, causing point and frameshift mutations as well as strand breaks, arrest of the cell cycle, and apoptosis [87]. Similar damage was shown for 4-HNE, which at increased concentration can cause mutations of GC to AT, interfere with DNA replication and transcriptions, and have genotoxic and cytotoxic roles. It has been shown that 4-HNE can cause mutations in codon 249 of the p53 tumor suppressor, which is a unique hotspot found in hepatocellular carcinoma [88]. Acrolein is another product of lipid peroxidation that has cancerogenic effects. Interchange DNA and protein DNA crosslinks formed as an acrolein effect were connected to cancers in several organs such as the liver, lung, and bladder [89,90]. Interestingly, acrolein has the ability to interact with the lysine residues of H4’s newly formed histones that interfere with chromatin assembly [91].

## 4. So It Begins: Obesity-Related Cell Transformation and Tumor Development

It was previously described how increased oxidative stress could be a cell-death trigger—the damaged cells undergo self-destruction as a way of damage control to limit the possible harmful effect on the surrounding cells and the organism as a whole. Various checkpoints and pathways involved in cell death consequential to oxidative stress are described elsewhere [29,92]. Oxidative stress, obesity-related as well as in general, has an important role in cell transformation and cancer development, with similar mechanisms determining the process (Figure 4). The transformed cells protect themselves by upregulating antioxidant enzymes and introducing metabolic adjustments to avoid cell death [17,85,93]. In fact, metabolic reprogramming is one of the prerogatives in the early-stage transition from normal to transformed cells since new demands are emerging, such as rapid proliferation [93]. While cancer cells successfully avoid cell death, oxidative stress is still constitutively increased in cancer relative to normal cells [94], and this is of particular significance for maintaining a high mutation rate and genomic instability as well [95]. With the critical oncogenes activated, protective mechanisms are propelled to limit and regulate the RONS level to the transformed cell advantage. Nrf2 is found upregulated in different types of cancer; the increased expression is considered essential to cancer cell proliferation and tumorigenesis and in correlation with tumor malignancy (Figure 4) [85,96]. DeNicola et al. reported that oncogenes K-Ras, B-Raf, and Myc regulate Nrf2 transcription, thus reducing oxidative stress [97]. Gain-of-function mutations of Nrf2 and inactivating mutations of Nrf2-inhibitor Keap1 have been reported in different types of cancer [27]. 

Reactive species can act as context-dependent regulators of oncogenic tyrosine kinases [98]. Accordingly, oncogenic tyrosine kinases modulate the activity of M2-type pyruvate kinase (PKM2) so that PKM2 promotes a redirection of glucose metabolites towards PPP, ultimately providing the cell with much-needed NADPH [99]. Since uncontrolled proliferation prerequisites a favorable redox milieu, the high level of NADPH can support both nucleotide synthesis and detoxifying redox systems [23,28]. As previously mentioned, both GSH and Trx systems depend on NADPH. Not surprisingly, Trx and GSH system proteins are often upregulated in cancer cells and essential in supporting tumor growth (Figure 4) [17,100]. RONS affect cell proliferation in numerous ways. One of them is through calcium release from intracellular depots, indirectly activating protein kinase C (PKC) [101].

Increased expression of NFκB is one of the standard features of transformed cancer cells (Figure 4). As one of the primary transcription regulators of IL-6, the hyperactivation of NFκB leads to an increase in IL-6. IL-6 is considered to be the critical factor connecting the state of chronic inflammation with tumor progression [102]. Increased IL-6 binds to receptor IL-6Rα, inducing the formation of an effector complex between IL-6, IL-6Rα, and IL-6Rβ/gp130, which further activates the JAK/STAT3 signaling pathway (Figure 4), leading to the transcription of STAT3 targeted genes. Along with the mutated genes for JAK in the occurrence of some tumors, IL-6 also contributes to JAK/STAT3 constitutive activation in tumor cells, thus enabling proliferation, survival, and suppression of antitumor immune response. Furthermore, STAT3 regulates the synthesis of IL-6, and IL-6 is also released by the immune cells in the tumor microenvironment. In normal cells, Nrf2 will activate to amortize the inflammation ROS by negative regulation of IL-6 and induction of PPARγ [103].

### 4.1. Examples of How Obesity-Driven Oxidative Stress Can Be a “Trump Card” to Cancer Development

Fat is stored primarily in subcutaneous adipose tissue (SAT) in healthy individuals. However, with long-term increased energy intake, leading to increased fat storage, the adipose tissue has a limited growth rate, thus, a limited fat storage capacity. Apart from excess accumulation of fat in SAT, obesity is accompanied by ectopic fat accumulation in visceral adipose tissue and the liver, skeletal muscles, pancreas, and heart [104,105]. The “out of place” fat tissue calls immune cells for help in an attempt to reestablish homeostasis. However, this chronic state of inflammatory response contributes more to organ damage than rescue. Furthermore, ectopic fat is considered a more significant risk factor in obesity and associated diseases outcome than subcutaneous fat accumulation [104,106,107,108]. This section will review research where adipose tissue accumulation promotes and supports the genetic and metabolic changes in neoplastic transformation.

In postmenopausal women, adipose tissue is the primary source of estrogen. Being exposed for a long time to high levels of estrogen may propose a risk factor and contribute to estrogen receptor-positive breast cancer development [109]. In obesity, the fat storage overload imposes an increased metabolic burden in adipocytes, consequently inducing a state of chronic inflammation, the release of transcription factors (e.g., NFκB, STAT3, HIF1α) and inflammatory cytokines (e.g., TNFα, Il-6, Il-8), subsequently leading to oxidative stress in breast fat tissue [85,110]. Therefore, in obesity, breast tissue is exposed to a surplus of estrogen, reactive species, and inflammation factors. Oxidative stress induces mutations in oncogenes and tumor suppressors, while alterations in growth and proliferation signaling in adipose tissue act as a support system for transformed cells. Further, the release of free fatty acids and inflammatory cytokines significantly contributes to remodeling the tumor microenvironment [111]. Mutations in breast cancer 1 (BRCA1) gene were discovered to be highly frequent in ovarian and breast cancer patients [112]. In the following investigations, BRCA1’s roles in cell cycle control, genome integrity maintenance, and gene transcription regulation were identified, yet the research on how it suppresses tumor occurrence is ongoing [113]. Relative to oxidative stress, Gorrini et al. [114] identified BRCA1 as an activator of Nrf2, contributing to Nrf2 stability and preventing oxidative stress-caused damage. Following, a tumor suppressor of high relevance to ovarian and breast tumor emergence is involved in response to oxidative stress as well, and loss-in-function contributes to redox disbalance leading to cancer formation. On the other hand, oxidative stress has a vital role in the recurrence of Her2-positive breast cancer [100]. Namely, the decrease in growth factor Her2 due to doxycycline treatment is followed by an increase in oxidative stress, and, at this point, Nrf2-driven metabolic reprogramming of the cancer cells are of pivotal significance for their survival and tumor progression of the dormant cells. 

Fat accumulation in the liver causes an inflammatory response. If this chronic response persists, the initial state of obesity might advance into organ damage, leading to nonalcoholic steatohepatosis (NASH), fibrosis, cirrhosis, and even hepatocellular carcinoma [115,116,117]. A meta-analysis from 2018 indicated that a BMI ≥ 30 is linked to twice the risk of HCC-associated mortality, while the same results were not observed for moderately obese with BMI ≥ 25 [118]. On the cellular level, excessive lipid droplets in the hepatocytes cause lipotoxicity, leading to cell death and immune cell activation. Cell death and inflammation further cause changes in the microenvironment of the liver, in some cases abetted by the genetic changes and cell signaling supportive of the neoplastic transformation [117]. All the while, these metabolic changes are accompanied by increased ROS concentrations. In addition, it has been reported that a high concentration of palmitate, associated with a high-fat diet, causes an increased glucose uptake and metabolism in vivo and in vitro and that these changes in glucose metabolism, much resembling the ones occurring in transformed, neoplastic cells [119], are dependent on peroxisomal oxidation of palmitate and H_2_O_2_ production [120]. Brahma et al. [121] recently described the sources and proposed mechanisms of how oxidative stress is connected to the development of hepatocellular carcinoma, pointing out that in some cases, the exact mechanisms of how obesity leads to HCC, without perceivable preceding liver damage, or why in some patients tissue damage is not followed by tumorigenesis remains to be determined. Beyond obesity, oxidative stress in the liver induced by other diseases is implicated in HCC development as well [122].

Pancreatic ductal adenocarcinoma is prevailingly a lethal disease: at early stages, the disease causes no symptoms and is considered practically undiscoverable until the tumor late progression, when it is basically untreatable. Strikingly, obesity more than doubles the risk of pancreatic cancer development [123]. Chronic pancreatitis and type 2 diabetes, both identified as predictable consequences of obesity, are also associated with the development of pancreatic cancer [124]. The exact mechanisms proposed as the linking factors between obesity and various cancer types are also suggested for obesity-related pancreatic cancer: chronic inflammation, insulin resistance, circulating lipids, and cytokines. Pro-inflammatory adipokines (MCP-1, IL-6, PAI-1), usually stimulated and increased in obesity, are closely associated with pancreatic tumors [49]. A suggested enhancer of carcinogenesis in pancreatic cancer is receptors of advanced glycation end products (RAGE), where AGE/RAGE signaling induces inflammation and generates reactive species, activating NFκB along the way [37]. Not long ago, it was pointed out that stress granules (SGs) have a substantial role in pancreatic cancer development and, in fact, that they are a compulsory component in the process of neoplasm formation [63]. SGs are assemblies of untranslating messenger ribonucleoprotein granules; they develop in response to different stressors, e.g., oxidative stress and inflammation, with a prominent role in determining the course of the stress response [125]. The animal-based study reported that the dependence of early-stage tumors upon SGs is more pronounced in stress conditions brought by obesity, with IGF1/IGF1R signalization identified as the principal mediator between the two conditions [63].

Protracted inflammation induces excessive RONS production, and increased RONS launch a pro-inflammatory response; essentially, chronic inflammation and reactive species form a positive feedback loop [126]. Excessive and unregulated production of reactive species over an extended period of time causes permanent harm to the cells. Considering all this, it would be reasonable to assume chronically inflamed tissue is more prone to succumb to neoplastic transformation. In the context of colorectal cancer (CRC), the first reports of how inflammatory bowel disease relates to cancer development date back almost a century ago [127]. The mechanistic description of how inflammation leads to colorectal cancer was previously depicted by Terzic et al. [128]. Several studies implied a hypercaloric diet, rich in fats and sugar, correlates with chronic colon inflammation and increased RONS [129,130,131]. Most frequently used is the chemically inducible murine model of colitis-associated cancer, which was developed by a combination of carcinogen drug injection azoxymethane and colitogen dextran sodium sulfate [132,133,134,135]. Through the induction of oncogenic pathways and inflammation, these two drugs induce normal crypt cells to transform, forming aberrant and divided crypts, with the ultimate formation of micro-tumors along the colon, at which the microadenomas location is a strain-specific trait [135]. The molecular features of this model are the activation of β catenin and Wnt signaling activity, enhanced inflammatory response (increased IL-6, TNFα, NFκB and a high number of inflammatory cells), increased activity of JAK/STAT3, as well as PI3K/Akt signaling pathway, and increased inducible nitric oxidase (iNOS). The described murine model closely simulates the human condition, with some discrepancies since additional activation mutations of KRAS oncogene and p53 tumor suppressor characterize human colon adenocarcinomas. 

Furthermore, intestinal microbiota, as a dynamic system, responds to food intake, and it has been reported a high-fat/high-sugar diet promotes dysbiosis and the advancement of pathogenic bacteria [136]. These pathogenic bacteria can further endorse inflammation and participate in supporting the formation of the tumor microenvironment. It has been suggested that intestinal inflammation promotes tumorigenesis through modifications in microbial content as well [137]. Some of the species identified as contributors to CRC carcinogenesis are *Fusobacterium nucleatum*, *Bacteroides fragilis*, and *Escherichia coli* [138]. In addition to oxidative stress being the consequence of inflammation, the microbiota can be a direct source of reactive species. For example, *Enterococcus faecalis*, another bacterial strain closely connected to CRC, in addition to inducing H_2_O_2_ production in macrophages, can also be a direct source of H_2_O_2_ [139]. The evidence of CRC-specific microbiome is accumulating [140,141], which led to suggesting the non-invasive stool sample analysis for the specific microbial strains as biomarkers and a screening test in CRC diagnostics. Knowledge of the importance of the pathogenic (and beneficial to host) species significantly extends to an effective therapeutic tool and manipulation of the microbiome to suppress tumorigenesis or cancer progression.

### 4.2. The Clandestine Connection between Obesity and Cancer

The process of cancer development in obese patients has been studied, and the main molecular pathways, primarily connected to adipokines circulation, oxidative stress, and inflammation, in different cancer types, have been identified. However, the relationship between obesity and cancer is not a linear one, and there is a lot more to be explained and investigated. For example, why do tumors develop in some obese patients and in some not? Or how does obesity in some periods of life affects the risk of tumor later in life? For example, childhood obesity can be a risk factor for some tumor types later in life [142,143,144], while premenopausal overweight women have a lower risk of developing breast cancer [145]. Are some tumor types, developed in tissues not directly involved in metabolism regulation, also triggered by obesity-induced adaptations? Are some micronutrients of particular importance for cancer development, and if so, is the cancer development in obesity the result of excess adipose tissue or nutritive deficiency? With this in mind, it appears the connecting mechanisms identified so far are just the tip of the iceberg, and this complex relationship has many faces yet to be recognized.

Recently, Sachdeva et al. [144] hypothesized how childhood obesity might be connected to glioblastoma, proposing systemic inflammatory cytokines as one of the contributors. Glioblastoma is the leading cause of brain cancer death; it accounts for about half of the malignant primary brain tumors, with the median survival if treated being 14.6 months [146]. Concerning 2005, although 2- and 3-year survival has increased among patients, 5-year survival remains unchanged and happens in only 3–4% of patients [147]. In more than a century of efforts, there have been only five drugs approved for treating glioblastoma, with little to no effect on patient survival rate [148]. In other words, glioblastoma remains a practically incurable malignant disease. 

A study by Howell et al. [149] indicated a positive correlation between genetically predicted childhood extreme obesity and all gliomas, glioblastoma included. Here, we would like to propose that systemic oxidative stress, along with chronic inflammation taking place in obesity, is a contributor to brain neoplasm development. A fructose-rich diet, alone and especially pronounced in combination with chronic stress, caused a decrease in antioxidant defense in the hypothalami of female rats [150]. Increased oxidative stress and inflammation in obese brain has been reported as well [151]. A decreased antioxidant defense would make the brain tissue more susceptible to oxidative damage and mutagenesis, thus leading to neoplastic transformation. Previous studies implied obesity, and BMI do not affect prognosis and survival in GBM patients [152,153]. However, an increased BMI in late adolescence increases the risk of glioblastoma development later on in life by four times [154], while being underweight in the early twenties decreases the probability of overall glioma development [155]. Despite possible decreased defense systems against oxidative stress before or at the time of occurrence of the tumor in brain tissue, leading to the transformation, it has been established glioblastoma patients have increased activity of the Trx system [156], in correlation with poor prognosis and higher resistance to therapy [157,158]. A highly active Trx system ensures the survival and proliferation of cancer cells in a variable, hypoxic, redox-imbalanced environment. Notwithstanding, the cancer cells are becoming dependent on the highly active Trx system, making it a perspective therapy target, which was further explored and discussed in more detail previously [17,159,160].

In obesity, both serum selenium (Se) and Se intake were implied as decreased [161,162,163]. Previously, low Se has been suggested as a risk factor for some cancers [164]. Despite high heterogeneity among the reported studies, a weight reduction increases Se antioxidant proteins, which suggests obesity indeed is a relevant debilitating factor in maintaining Se homeostasis [165]. An alluring hypothesis is that Se deficiency observed in obesity is one of the important culprits in setting the spark of the event cascade related to thyroid cancerogenesis. In thyrocyte follicular cells, thyroid peroxidases use H_2_O_2_ as the oxidizing agent for the iodine incorporation into thyroglobulin, which could be considered an essential step in thyroid hormone synthesis. The delicate balance between H_2_O_2_ production and reactive oxygen species neutralization in thyrocytes is of utmost importance. Although H_2_O_2_ is not a direct threat to DNA, the derived hydroxyl radicals are quite efficient damage-inducing oxidizing agents. Increased oxidative stress and DNA damage that follows are considered to be one of the initial steps in thyroid cancer tumorigenesis [166,167]. Multiple studies reported decreased antioxidative defense, accompanied by increased oxidative stress in tumor tissue and serum of thyroid cancer patients [168]. Interestingly, the increased concentration of reactive species was localized and more pronounced at the edge of the tumor compared to the tumor core and normal thyroid tissue [169]. Importantly, the detected oxidative stress corresponded to more malignant tumors [169]. The main enzyme catalysts of H_2_O_2_ are Prx (functionally Trx system dependent), GPx, and catalase. Studies have reported a decreased content and/or activity of TrxR1, GPx, and catalase in the cancer tissue and serum of thyroid cancer patients [170,171]. As suggested by some [172], the decrease of these crucial detoxifying selenoproteins, TrxR1 and GPx, is possibly a consequence of the Se decrease in thyroid cancer patients.

## 5. Discussion

In this manuscript, we explored the up-to-date literature on the molecular connections between obesity and cancer, how one disease leads to another, and suggested oxidative stress is an important protagonist in this progression. It is important to draw attention to how the state of obesity can insidiously change the cellular microenvironment, causing the cells to transform and form malignancies we might not be able to detect until it’s too late. 

Investigating Western diet-induced pathology, and non-communicable diseases association, is most often approached through BMI-defined obesity. However, BMI, as well as waist circumference and waist-to-hip ratio, are not without limitations in assessing whether a person is negatively affected by a mass increase or a diet. For example, some professional sportsmen can be classified as obese, according to the BMI, but their mass increase can be on account of the muscle mass, not adipose tissue, and as such, they are not at higher risk for most non-communicable diseases. On the other hand, there are “normal-weight obese” or “metabolically obese” [173,174] individuals with a high percentage of body fat, visceral obesity, low percentage of muscle mass, insulin resistance, and dyslipidemia, and generally, a higher risk of developing metabolic disease and possibly, related ones, including cancer. High body fat and impaired adipose tissue homeostasis are considered more reliable indicators of insulin resistance and attributed complications than anthropometric parameters. The lack of straightforward criteria makes it difficult to truly grasp the effect oxidative stress, resulting from metabolic stress, has on cancer development. As striking as it might seem, with the data as limited as they are, and with all previously considered, this implies the prognosis of cancer might be an even more probable outcome of (metabolic) obesity than the BMI-based studies imply.

Treating cancer in obesity is particularly challenging [175,176]. A cause of this is that doses and treatment regimens are developed on a normal weight population, taking into consideration body surface area. Recommendations are that special care should be taken when calculating the doses for overweight and obese patients, taking into account their body weight. More often than not, obese patients with cancer have an increased poor prognosis, despite the treatment success in the general population [177]. Adipose tissue can be of particular nuisance in chemotherapy, as it can sequester some of the therapeutics [178] or provide cancer cells with alternative survival strategies [179]. The tumor microenvironment in obese patients differs from normal weight patients, meaning the research of therapeutic efficacy may not apply to this “unknown” setting. However, it has been shown in some types of cancers and particular therapeutic modalities, obese patients have a better response to therapy, compared to normal-weight patients, by experiencing decreased toxicity to chemotherapeutics, an improved response to a combination of radiation and chemo, and enhanced efficacy of immunotherapy [177]. An obvious question imposes itself—how does weight reduction affect cancer treatment and the disease outcome? Despite some discrepancies in reported studies ascribed to differences in weight loss programs, in most cases, reduction of weight through balanced nutrition, with moderate caloric reduction, accompanied by structured and guided physical activity and behavioral changes, reduces the complications of cancer treatment, improves the quality of life and survival outcome [176].

Apart from cancer being a preventable, possibly deadly, outcome of obesity, affecting the quality of life for both patients and their loved ones, not insignificant is the economic burden obesity brings to society. Overweight and obesity burden was estimated at 2.19% of the global growth domestic product (GDP), with projections that by 2060, taking into account current rates, this will make up to 3.29% of global GDP [180]. Reducing the current rates of overweight and obesity by 5% per year could translate into a global annual reduction in costs of illness by $429 billion [180].

A weight reduction ≥ 5% of total body mass reduces the obesity-related cancer risk for breast, endometrial, and colorectal cancer in postmenopausal women [181,182,183]. Principally, weight reduction is advised to be accomplished through adopting healthy life habits, balanced nutrition with moderate caloric restriction, and increased physical activity. For example, moderate caloric restriction is shown to decrease markers of oxidative stress and partially improve the oxidative status of the patients [184]. Healthy dietary regimes, such as the Mediterranean diet, have been suggested as well for improving health parameters by reducing inflammation and oxidative stress in obesity, thus reducing the risk of cancer [185]. Physical activity is known to be a “two-edged sword” regarding oxidative stress—in moderation and with progressive improvement, it can ameliorate oxidative stress and boost the immune system, but excessive physical activity (and this can be rather individual what excessive is) can be the source of oxidative stress and promote inflammation [186,187,188]. Bariatric surgery in obese patients reduces the risk of obesity-related cancers as well [189]. Although limited, some pharmaceutical solutions for treating obesity with remarkable improvement in health parameters do exist [190]. Other pharmaceuticals that do not directly target obesity are being investigated, but the evidence of cancer risk reduction is quite weak so far [191].

Another important issue is society’s perception of the link between obesity and carcinogenesis. There are studies showing that awareness of obesity as a risk factor for cancer is suboptimal, and obese people often do not have heightened risk perceptions [2,192,193]. In the study that assessed average cancer risk versus personal risk based on personal characteristics and behaviors, only 52% of respondents correctly identified obesity as a risk factor for cancer [194,195]. The results of the study by Silverman et al. [195] showed that the likelihood of identifying obesity as a risk factor for cancer was significantly lower compared with having a family history of cancer, while Burkbauer et al. [196] showed that knowledge of the association between obesity and breast cancer risk was associated with willingness to participate in a weight loss intervention. It appears that future prospects rest on the development of better education and communication tools to improve awareness of the link between obesity and cancer. Such an approach is likely to improve the adoption of healthy lifestyles, especially among high-risk patients.

To answer the question regarding obesity being the ‘radical trigger’ to cancer, the conditions of oxidative stress and inflammation taking place in obesity certainly can propose a threat to human health and result in cancer in some cases. The hypothesis that started as a series of correlation studies is starting to unveil the molecular pathways at play. The relationship between obesity and cancer is quite complex and, in some circumstances, a challenge to explain straightforwardly, especially when there is a time gap between the two states. The origin of cancer is convoluted, and, in all likelihood, there is not one thing that causes it. However, the identification and understanding of the culprits that can be dealt with can provide opportunities to reduce the risk and perhaps even prevent cancer. The prevention of chronic obesity should be considered an important part of cancer prevention. When compared to the overall cost of obesity and related disease, from an economic point of view, it seems reasonable to invest in education about healthy lifestyles and solutions for dealing with obesity before the problem worsens. From the humane point of view, curing obesity offers a choice to fight a battle one may win.

## Figures and Tables

**Figure 1 ijms-24-08452-f001:**
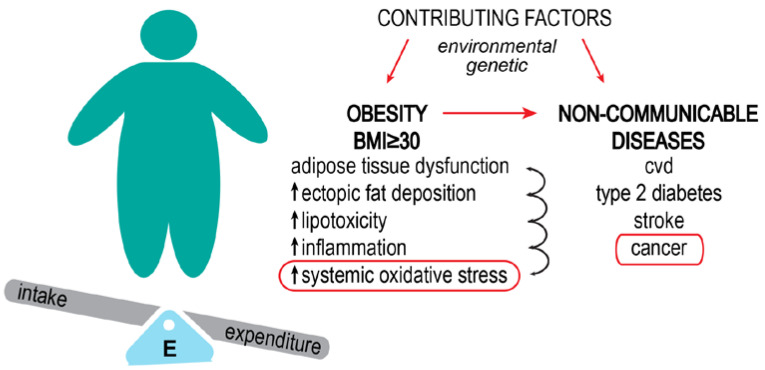
Obesity increases the risk of developing non-communicable diseases. Obesity results from the interaction of environmental factors (high food intake, low physical activity) and genetic factors (some gene variants are thought to play a role in the development of the obese phenotype). The basis of obesity is the imbalance between energy intake and expenditure, which leads to increased body fat percentage and body mass index (BMI). Perpetual increased energy leads to metabolic dysfunctions. Systemic oxidative stress is a consequence of this impaired metabolism. Obesity-related cancer types are considered preventable non-communicable diseases.

**Figure 2 ijms-24-08452-f002:**
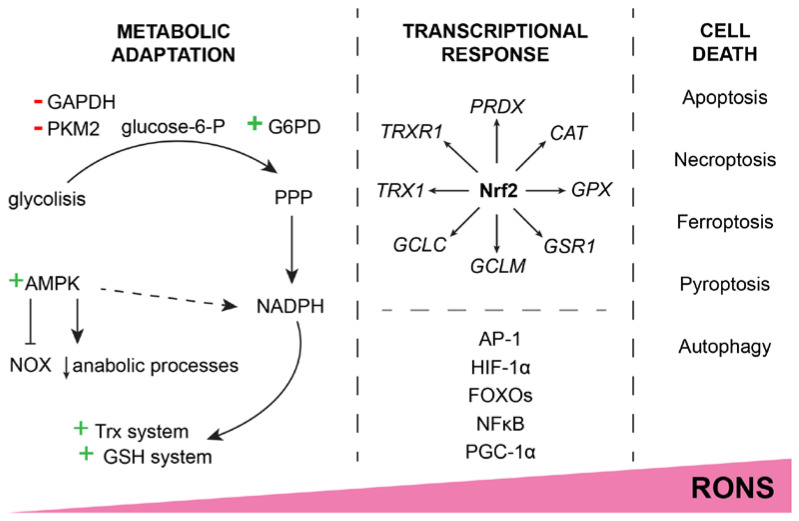
Increase in RONS concentration causes an adaptive response or cell death. Moderate redox imbalance leads to metabolic adaptation by shifting glycolysis products to the PPP, inhibiting RONS generators (e.g., NOX), and generally inhibiting anabolic processes with the main purpose of supplying NADPH to the Trx and GSH detoxification systems. A sustained increase in RONS leads to a transcriptionally programmed response. At the center of the transcription factors that respond to the increase in RONS is Nrf2, but other factors (AP-1, FOXOs, NFκB, etc.) also contribute to the defense. Should the RONS continue to cause cell damage, the cell eventually triggers some of the cell death pathways to maintain the integrity of the surrounding tissue. The exact mechanism of triggered cell death depends on the context of oxidative stress.

**Figure 3 ijms-24-08452-f003:**
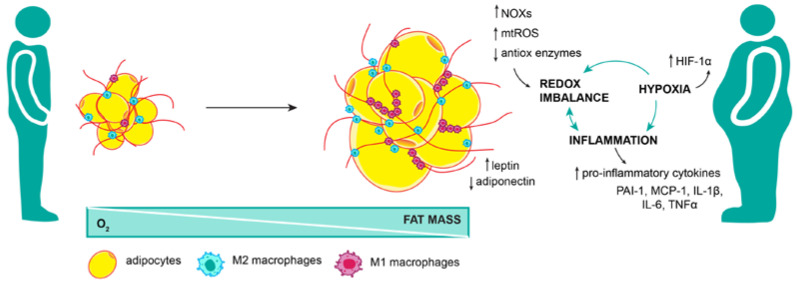
Adipose tissue dysfunction brings oxidative stress. Adipose tissue expansion in obesity is not harmonized with angiogenesis; therefore, as fat mass increase, portions of the tissue are cut off from the oxygen supply. This state of hypoxia further exacerbates inflammation and redox imbalance in adipose tissue. Pro-inflammatory M1 macrophages, along with increased NOX activity, mitochondrial ROS production, and decreased expression of antioxidant enzymes, lead to systemic oxidative stress not limited to adipose tissue.

**Figure 4 ijms-24-08452-f004:**
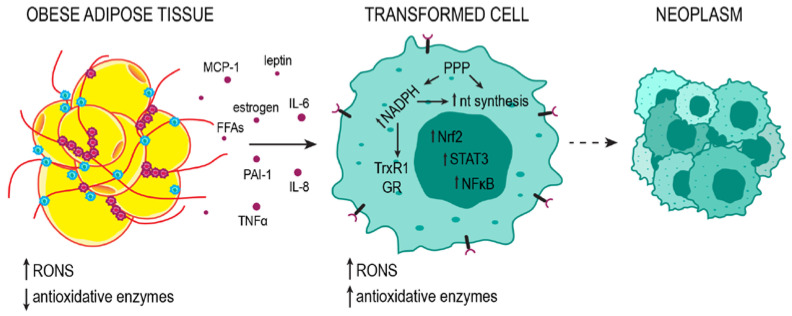
Adipose tissue promotes the neoplastic transformation of the cells. Adipose tissue releases pro-inflammatory cytokines, hormones, and free fatty acids (FFAs) into the bloodstream. The resulting systemic oxidative stress and inflammation promote mutagenesis in oncogenes and tumor suppressor genes, contributing to cell transformation. In transformed cells, high metabolic activity leads to high RONS concentrations. To avoid cell death, the cell increases the production of antioxidant enzymes; nevertheless, RONS levels are relatively high and support genomic instability. Metabolism subordinate to rapid growth and proliferation favors a reducing environment in which NADPH, with PPP as the major source, can be used for the components of the Trx and GSH detoxification systems (TrxR1 and GR) and for nucleotide (nt) synthesis. Uncontrolled proliferation leads to the formation of neoplasms.

## Data Availability

Not applicable.

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
