# Peer review of "Oxidative Stress Linking Obesity and Cancer: Is Obesity a ‘Radical Trigger’ to Cancer?"

_ijms, 2023, doi:10.3390/ijms24098452_

Round 1

Reviewer 1 Report

The article Oxidative Stress in Obesity and Cancer: Is Obesity a ‘Radical Trigger’ to Cancer? – underline the close link between vast pathological fields due to chronic oxidative stress and inflammation caused by metabolic adaptations in obese individuals. The connection between obesity and cancer is complex, but identifying culprits can help reduce the risk and prevent it. A healthy lifestyle that includes hypocaloric, balanced nutrition, and structured physical activity can help alleviate this burden. However, treating both Obesity and Cancer simultaneously poses a challenge. Based on my analysis, here are some recommendations to improve the article:

1. Clarify the title: The title of the article, "Is obesity the 'radical trigger' to Cancer?" could be improved by adding a clear indication that the article is discussing the correlation between Obesity and Cancer. This would make it easier for readers to know what to expect from the article.

2. Provide more context: The article could benefit from providing more context on the topic of Obesity and Cancer. For instance, it could briefly explain what cancer is, how it develops, and its risk factors.

3. Define key terms: The article uses some technical terms such as "chronic oxidative stress" and "systemic inflammation" without defining them. Defining these terms would make the article more accessible to readers who may not be familiar with them.

4. Provide more examples: The article could provide more concrete examples to illustrate the link between Obesity and Cancer. For instance, it could discuss specific types of Cancer that are more common in obese individuals, or provide statistics on the incidence of Cancer in obese populations.

5. Discuss limitations: The article could benefit from discussing the limitations of the research on Obesity and Cancer. For instance, it could mention the difficulty of establishing causality in observational studies, or the fact that different types of Cancer may have different risk factors.

6. Offer more specific recommendations: The article mentions the importance of preventing chronic Obesity as a means of cancer prevention but does not offer specific recommendations for how to do so. Providing more specific recommendations, such as increasing physical activity or reducing sugar intake, would make the article more actionable for readers.

7. It will be of interest to understand the population's perception regarding the link between obesity and cancer. Also, future research directions.

8. Objectives of the article will help understand the direction of the article.

9. Why not a more explicit structure of the article to comprehend the entire picture of such a vast topic?

10. How were the studies included in the references selected? Why not a systematic review of such a complex subject?

The quality of the English Language is fine for a non-native speaker. It is clear, and only minor corrections will be necessary by proofreading.

Author Response

  1. Clarify the title: The title of the article, "Is obesity the 'radical trigger' to Cancer?" could be improved by adding a clear indication that the article is discussing the correlation between Obesity and Cancer. This would make it easier for readers to know what to expect from the article.

Answer: Previous title of the manuscript “Oxidative Stress in Obesity and Cancer: Is Obesity a ‘Radical Trigger’ to Cancer?” is now modified according to the reviewer’s suggestion. The new title: “Oxidative Stress Linking Obesity and Cancer: Is Obesity a ‘Radical Trigger’ to Cancer?”, now more clearly indicates that the manuscript is dedicated to oxidative stress as a factor connecting these diseases and implies (‘radical trigger’ is an intended word-play) that oxidative stress has an important role in cancer development.

  1. Provide more context: The article could benefit from providing more context on the topic of Obesity and Cancer. For instance, it could briefly explain what cancer is, how it develops, and its risk factors.

Answer: We appreciate the reviewer's comment. A brief explanation of what cancer is, how it develops, and what the risk factors are, including overweight and obesity, is already discussed in the introduction (lines 50-67 in the revised version). Since reviewer 2 suggests that "the information in lines 58-75 (in the revised version, this is the paragraph 50-67) should be shortened" we didn't have much leeway to meet the requirements of both reviewers and avoid a conflict.

  1. Define key terms: The article uses some technical terms such as "chronic oxidative stress" and "systemic inflammation" without defining them. Defining these terms would make the article more accessible to readers who may not be familiar with them.

Answer: A general definition of the term "oxidative stress" and reactive oxygen and nitrogen species is already given (lines 86-96 in the revised version). Since the term "chronic oxidative stress" is mentioned only twice in the text (lines 4 and 657 in the previous version), we decided to omit the word "chronic" in order not to confuse the reader. Similarly, the term "systemic inflammation" is used only once (line 657 of the previous version), and because the term "inflammation" is broad enough to be used alone, we decided to omit the word "systemic."

  1. Provide more examples: The article could provide more concrete examples to illustrate the link between Obesity and Cancer. For instance, it could discuss specific types of Cancer that are more common in obese individuals, or provide statistics on the incidence of Cancer in obese populations.

Answer: We mentioned concrete examples of some specific cancer types more common in obese individuals, in the Introduction (lines 57-60 in the revised version). Further, the whole section 4. So it begins: obesity-related cell transformation and tumor development, is dedicated to concrete examples of cancer types, relatively more frequent in obese patients, where the oxidative stress linking the progression from obesity to cancer was well covered in the literature. For example, (lines 445-447 in the rev.ver.). “A meta-analysis from 2018 indicated that a BMI≥30 is linked to twice the risk of HCC-associated mortality, while the same results were not observed for moderately obese with BMI≥25 [118].” Or, “A study by Howell et al. [146] indicated a positive correlation between genetically predicted childhood extreme obesity and all gliomas, glioblastoma included. ” (lines 545-547, rev. ver.).

  1. Discuss limitations: The article could benefit from discussing the limitations of the research on Obesity and Cancer. For instance, it could mention the difficulty of establishing causality in observational studies, or the fact that different types of Cancer may have different risk factors.

Answer: According to the reviewer's suggestion, we added the limitations in the research of Obesity and Cancer lines (596-612).

  1. Offer more specific recommendations: The article mentions the importance of preventing chronic Obesity as a means of cancer prevention but does not offer specific recommendations for how to do so. Providing more specific recommendations, such as increasing physical activity or reducing sugar intake, would make the article more actionable for readers.

Answer: We appreciate the reviewer’s comment. The specific strategies of weight reduction have already been discussed in the Discussion section (lines 640-656, rev.ver.): "Weight reduction ≥ 5% of total body mass reduces the obesity-related cancer risk for breast, endometrial, and colorectal cancer in postmenopausal women [181-183]. Principally, weight reduction is advised to be accomplished through adopting healthy life habits, balanced nutrition with moderate caloric restriction, and increased physical activity. For example, moderate caloric restriction is shown to decrease markers of oxidative stress and partially improve the oxidative status of the patients [184]. Healthy dietary regimes, such as the Mediterranean diet, have been suggested as well for improving health parameters, by reducing inflammation and oxidative stress in obesity, thus reducing the risk of cancer [185]. Physical activity is known to be a “two-edged sword” regarding oxidative stress – in moderation, and with progressive improvement it can ameliorate oxidative stress and boost the immune system, but excessive physical activity (and this can be rather individual what excessive is) can be the source of oxidative stress and promote inflammation [186-188]. Bariatric surgery in obese patients reduces the risk of obesity-related cancers as well [189]. Although limited, some pharmaceutical solutions for treating obesity, with remarkable improvement of health parameters do exist [190,191]. Other pharmaceuticals, not targeting obesity directly, are being investigated, but the evidence of cancer risk reduction is quite weak so far [192]."

Since the second reviewer suggested that we shorten the discussion because it was too long, this paragraph was not elaborated on.

  1. It will be of interest to understand the population's perception regarding the link between obesity and cancer. Also, future research directions.

Answer: We agree with the reviewer that popular perception is an important aspect to be investigated in developing strategies for addressing the prevention of cancer through treating obesity. Therefore, text on this topic has been added to the manuscript (lines 657-669, rev.ver.).

  1. Objectives of the article will help understand the direction of the article.

Answer: Objectives of the article have been pointed out in the Introduction (lines 69-76, rev. ver.).

  1. Why not a more explicit structure of the article to comprehend the entire picture of such a vast topic?

Answer: We understand the reviewer's concerns, but we are convinced that, regardless of the structure, the manuscript covers all the important aspects of the topic. We have described what oxidative stress is and what kind of cell response can be (Section 2). We have further described how molecular pathological mechanisms that occur in obesity cause oxidative stress and inflammation, referencing how this condition leads to certain types of cancer (Section 3). In Section 4, we described specific examples and mechanisms by which oxidative stress in obesity leads to specific cancers that are more common in obese patients. Finally, the less formal headings and subheadings are intended to be interesting and engaging to the reader.

  1. How were the studies included in the references selected? Why not a systematic review of such a complex subject?

Answer: This Manuscript was not meant to be a systemic review, but a contribution to investigating how obesity is connected to cancer through oxidative stress. References were selected based on their relevance and impact on the field.

Reviewer 2 Report

The topic of the work taken up by the authors is very interesting and concerns contemporary health problems, the work is generally well written, however, I have a few comments: 1. The introduction is too long, the authors focus on information that is not relevant to the topic of the manuscript: - information from the first paragraph is too obvious to include in the publication - information about COVID-19 should be shortened to 1 sentence, literally that the pandemic had an impact on the increase in the number of obese people - information from lines 58-75 should be shortened - the authors should emphasize in the introduction that obesity is a factor in the development of cancer 2. Metabolic pathology in obesity brings systemic havoc: - in this chapter, the authors describe only part of the mechanisms related to metabolic complications, this part should be shortened and added information on nitrosative stress and glycooxidation in obesity, besides, these changes and the intensity of the oxidative control depend on gender 3. So it begins: obesity-related cell transformation and tumor development: - oxidative stress is involved in the development of tumors not only in obese people 4. the discussion is too long, it should be shortened, and the authors should focus mainly on the topic of the manuscript 5. in general, a paragraph should be added in the paper about defense mechanisms against oxidative hepatitis, disruption of the antioxidant barrier in obese people and total antioxidant/oxidative potential, which accurately describes redox homeostasis

Author Response

  1. The introduction is too long, the authors focus on information that is not relevant to the topic of the manuscript:

- information from the first paragraph is too obvious to include in the publication 

Answer: The first paragraph is now significantly shortened.

- information about COVID-19 should be shortened to 1 sentence, literally that the pandemic had an impact on the increase in the number of obese people 

Answer: We appreciate the reviewer's comment. The paragraph is now shortened. However, to our knowledge, the COVID -19 implications for obesity (and via obesity for other non-communicable diseases) have not been discussed in detail. We believe that attention should be drawn to the fact that the COVID -19 pandemic and lockdown may have exacerbated obesity rates, and that is the reason why we have given some more context to the few studies that have been done on this topic to date.

- information from lines 58-75 should be shortened 

Answer: Since the other reviewer suggests that the brief explanation of what cancer is, how it develops, and the risk factors (all of which are mentioned in the introduction - lines 58-75, in the current version in lines 50-67) should be elaborated on to provide more context, we didn't have much room to accommodate the requirements of both reviewers.

- the authors should emphasize in the introduction that obesity is a factor in the development of cancer 

Answer: According to reviewer's suggestion, we added the statement in the Introduction (line 68).

  1. Metabolic pathology in obesity brings systemic havoc:

- in this chapter, the authors describe only part of the mechanisms related to metabolic complications, this part should be shortened and added information on nitrosative stress and glycooxidation in obesity, besides, these changes and the intensity of the oxidative control depend on gender 

Answer: According to the reviewer’s suggestion, we made the adjustments in section 3 (lines 206-216 and 287-295).

  1. So it begins: obesity-related cell transformation and tumor development:

- oxidative stress is involved in the development of tumors not only in obese people 

Answer: According to the reviewer’s suggestion, we added a statement in section 4 (line 352), where we made it clear oxidative stress in general contributes to cancer development. In the introduction of section 4, we described how oxidative stress in general contributes to tumor development. Previously, in the conclusion of section 2, we mentioned how oxidative stress could contribute to the neoplastic transformation of the cell (line 170).

  1. the discussion is too long, it should be shortened, and the authors should focus mainly on the topic of the manuscript

Answer: According to the reviewer’s suggestion, we shortened the discussion. However, due to suggestions made by the other reviewer, we decided to leave some parts on-topic in a broad sense.

  1. in general, a paragraph should be added in the paper about defense mechanisms against oxidative hepatitis, disruption of the antioxidant barrier in obese people and total antioxidant/oxidative potential, which accurately describes redox homeostasis

Answer: We thank the reviewer for bringing our attention to this. In accordance with the suggestion, we added a statement in section 4 (line 460), with the appropriate reference discussing this problematic in more detail. Due the fact that the subject of the paper is the role of obesity-induced oxidative stress in cancer development and not oxidative stress in general, we would restrain ourselves from further elaborating on the matter. The disruption of the antioxidant barrier and the disruption of redox homeostasis in obese people have previously been discussed in Sections 3 and 4.

Round 2

Reviewer 1 Report

The authors have addressed the comments and suggestions in their responses to the previous evaluation report. The title of the article has been modified to "Oxidative Stress Linking Obesity and Cancer: Is Obesity a ‘Radical Trigger’ to Cancer?", which now more clearly indicates that the manuscript discusses the connection between oxidative stress, obesity, and cancer. The authors have also provided additional context on the topic of obesity and cancer in the introduction section, including a brief explanation of what cancer is, how it develops, and the risk factors associated with it. Generally, considering constraints and other reviewers' recommendations, as indicated by authors in their responses, I conclude for Accept in present form

No detected issues.

Reviewer 2 Report

The manuscript may be published in its present form